# The interaction of DNA repair factors ASCC2 and ASCC3 is affected by somatic cancer mutations

Junqiao Jia [1], Eva Absmeier[1,5], Nicole Holton[1], Agnieszka J. Pietrzyk-Brzezinska[1,6], Philipp Hackert[2], Katherine E. Bohnsack [2], Markus T. Bohnsack[2,3] & Markus C. Wahl [1,4✉]

The ASCC3 subunit of the activating signal co-integrator complex is a dual-cassette Ski2-like nucleic acid helicase that provides single-stranded DNA for alkylation damage repair by the α-ketoglutarate-dependent dioxygenase AlkBH3. Other ASCC components integrate ASCC3/ AlkBH3 into a complex DNA repair pathway. We mapped and structurally analyzed interacting ASCC2 and ASCC3 regions. The ASCC3 fragment comprises a central helical domain and terminal, extended arms that clasp the compact ASCC2 unit. ASCC2–ASCC3 interfaces are evolutionarily highly conserved and comprise a large number of residues affected by somatic cancer mutations. We quantified contributions of protein regions to the ASCC2–ASCC3 interaction, observing that changes found in cancers lead to reduced ASCC2–ASCC3 affinity. Functional dissection of ASCC3 revealed similar organization and regulation as in the spliceosomal RNA helicase Brr2. Our results delineate functional regions in an important DNA repair complex and suggest possible molecular disease principles.

[1] Laboratory of Structural Biochemistry, Freie Universität Berlin, D-14195 Berlin, Germany. [2] Department of Molecular Biology, University Medical Centre Göttingen, Göttingen, Germany. [3] Göttingen Center for Molecular Biosciences, Georg-August-Universität, Göttingen, Germany. [4] Helmholtz-Zentrum Berlin für Materialien und Energie, Macromolecular Crystallography, D-12489 Berlin, Germany. [5] Present address: MRC Laboratory of Molecular Biology, Cambridge Biomedical Campus, Cambridge CB2 0QH, UK. [6] Present address: Institute of Molecular and Industrial Biotechnology, Faculty of Biotechnology and Food Sciences, Lodz University of Technology, Lodz 90-924, Poland. ✉email: markus.wahl@fu-berlin.de

The human genome is constantly under assault by endogenous or exogenous DNA damaging agents. To ward off these insults, cells have evolved systems to recognize DNA damage, signal its presence and initiate repair processes[1]. Among the diverse repair mechanisms, direct DNA repair processes represent efficient means to revert chemical changes to DNA and involve enzymes such as photolyases, alkyl-transferases or dioxygenases[2,3]. *Escherichia coli* α-ketoglutarate-dependent dioxygenase, AlkB, homologs (AlkBH's) are one class of important DNA repair factors that reverse N-alkyl lesions[4]. Among the nine identified AlkBH enzymes in human, AlkBH2 and AlkBH3 dealkylate 1-methyl adenosine and 3-methyl cytosine[5,6].

Several lines of evidence implicate the human activating signal co-integrator complex (ASCC) in AlkBH3-mediated DNA repair. ASCC is composed of four subunits, ASCC1, ASCC2, ASCC3, and ASC1/TRIP4 (refs. [7,8]). ASCC3 is the largest subunit of ASCC and was characterized as a DNA helicase that unwinds DNA by translocating on one strand in 3'-to-5' direction[8]. The enzyme is thought to provide single-stranded DNA as a substrate for de-alkylation repair by AlkBH3 (ref. [8]). ASCC and alkylated nucleotides co-localize at nuclear foci upon alkylation damage stress, dependent on a coupling of ubiquitin conjugation to ER degradation (CUE) domain in ASCC2, which links DNA alkylation damage repair to upstream ubiquitin signaling via the RING finger protein 113A[9]. ASCC1 is cleared from these foci upon DNA alkylation damage and knockout of ASCC1 leads to loss of ASCC2 from the nuclear foci and increased cellular sensitivity to alkylating insults[10].

Both ASCC2 and ASCC3 have been linked to various human diseases. ASCC2 is upregulated in patients with rheumatoid arthritis[11], and ASCC3 is upregulated in peripheral blood mononuclear cells from patients with lung cancer[12,13]. A role of ASCC3 in cancer development or progression is also suggested by the observation that knockdown of ASCC3 in a prostate cancer cell line suppresses cell proliferation[8]. Moreover, the Catalog Of Somatic Mutations In Cancer (COSMIC v91, release date 07. April 2020; https://cancer.sanger.ac.uk/cosmic)[14] lists 223 and 652 somatic nonsense/missense/frame shift mutations in the *ascc2* and *ascc3* genes, respectively (822 and 2197 tested human cancer samples, respectively). As cancers can be caused by the acquisition of somatic mutations[15], the large number of somatic *ascc2* and *ascc3* mutations observed in cancer samples might be linked to the roles of ASCC/AlkBH3 in sensing and repairing alkylated DNA lesions. Moreover, chemotherapy is one of the most widely used cancer treatments and causes various DNA lesions, including alkylation damage[16]. The functional interplay of ASCC and AlkBH3 in DNA de-alkylation repair may thus also have repercussions for cancer chemotherapy[9].

Apart from DNA de-alkylation repair, ASCC has been implicated in a number of other cellular processes. The complex was originally discovered as a co-activator of nuclear hormone receptors[17]. In addition, ASCC3 has been implicated in viral defense mechanisms[18–21]. Finally, a genome-wide CRISPRi screen showed that ASCC2 and ASCC3 are the two most potent modifiers of cell fitness in the presence of a translation inhibitor, suggesting that the proteins affect stalled ribosomes[22]. Indeed, ASCC3 has been suggested to resolve stalled ribosomes on poly-A sequences in a ribosome quality control pathway[23–25]. Likewise, its yeast homolog, Slh1p, has been implicated in ribosome-associated quality control to prevent the accumulation of aberrant proteins[23,26,27] and in non-functional ribosomal RNA decay[28].

As a member of the Ski2-like family of superfamily (SF) 2 nucleic acid helicases, ASCC3 shows striking resemblance to the pre-mRNA splicing factor Brr2 (sequence identity of 41/33% between human ASCC3 and human/yeast Brr2), an ATP-dependent RNA helicase that is essential for spliceosome

activation, catalysis and disassembly[29]. Based on the well-documented structural organization of Brr2 (refs. [30–33]), ASCC3 is predicted to encompass a large N-terminal region (NTR; residues 1–400) and a helicase region (HR; residues 401–2202) containing a tandem array of two Ski2-like helicase cassettes. Each helicase cassette is predicted to encompass two RecA-like domains, a winged-helix domain and a Sec63 homology unit; the latter elements comprise sequential helical bundle or "ratchet", helix-loop-helix and immunoglobulin-like domains. Brr2 is a subunit of the U4/U6•U5 tri-snRNP[34,35]. ATPase/helicase activities of Brr2 are tightly regulated both intrinsically and by interacting proteins[33,36–39]. The high sequence similarity between ASCC3 and Brr2, and the observation that ASCC3, like Brr2, interacts with a number of other proteins, suggest that ASCC3 might be regulated by similar principles.

Structural knowledge regarding ASCC subunits and their various interactions will be instrumental in further delineating the molecular mechanisms by which these proteins participate in diverse cellular functions and how they are linked to human diseases. While some interacting regions among ASCC subunits have been broadly mapped[9,10], the structural basis of these interactions is presently unclear. Currently, only an unpublished NMR structure of the isolated ASCC2 CUE domain (PDB ID 2DI0) is known.

Here, we mapped precise interacting regions of the two largest ASCC subunits, ASCC2 and ASCC3, and determined a crystal structure of an ASCC2–ASCC3 complex comprising these portions. Guided by the structure, we delineated segments and residues that are critical for the ASCC2–ASCC3 interaction, which include a large number of residues that are altered by somatic cancer mutations. We also further investigated ATPase/helicase activities of ASCC3, and how they may be impacted by ASCC2.

## Results

**Experimental definition of a stable, minimal ASCC2–ASCC3 complex.** Full-length ASCC2 (ASCC2$^{FL}$) and two fragments of ASCC3, encompassing the NTR (ASCC3$^{NTR}$; residues 1–400) and helicase region (ASCC3$^{HR}$; residues 401-2202), were obtained by recombinant expression in insect cells. Analytical size exclusion chromatography (SEC) showed that ASCC2 interacts with ASCC3$^{NTR}$ (Fig. 1a) but not with ASCC3$^{HR}$ (Fig. 1b), consistent with previous reports[9]. The complex assembled from ASCC2$^{FL}$ and ASCC3$^{NTR}$ failed to crystallize. To remove putatively flexible regions that may hinder crystallization, we subjected the ASCC2$^{FL}$–ASCC3$^{NTR}$ complex to limited proteolysis and mapped stable fragments by mass spectrometric fingerprinting. Elastase digestion gave rise to an approximately 50 kDa fragment of ASCC2, containing the first 434 residues (ASCC2$^{1–434}$), and an N-terminal, 207-residue fragment of ASCC3 (ASCC3$^{1–207}$), which maintained stable interaction in SEC (Fig. 1c). The ASCC2$^{1–434}$–ASCC3$^{1–207}$ complex was produced by recombinant co-expression in insect cells, purified and yielded diffracting crystals.

**Structure analysis and overall architecture.** The crystal structure of the ASCC2$^{1–434}$-ASCC3$^{1–207}$ complex was determined via the single wavelength anomalous dispersion (SAD) approach, using a complex reconstituted by SEC from seleno-methionine (SeMet)-derivatized ASCC2$^{1–434}$ produced in *E. coli* and ASCC3$^{1–207}$ produced in insect cells (Supplementary Fig. 1). The structure was refined at 2.7 Å resolution to $R_{work}/R_{free}$ values of 20.4/24.7% with good stereochemistry (Supplementary Table 1). An asymmetric unit of the crystals contains one ASCC2$^{1–434}$–ASCC3$^{1–207}$ complex. In the final model, we traced residues 2–408 of ASCC2$^{1–434}$

with a gap between residues 216–226 representing a flexible loop. The ASCC3$^{1-207}$ model comprises residues 1–186 without gaps and two additional residues at the N-terminus that were retained after tag cleavage.

Within the complex, the structure of ASCC2$^{1-434}$ exhibits an irregularly structured N-terminal region (residues 2–50), and two similarly structured, helical sub-domains (sub-domain A, residues 51–214; sub-domain B, residues 241–406; Fig. 1d, e). Both

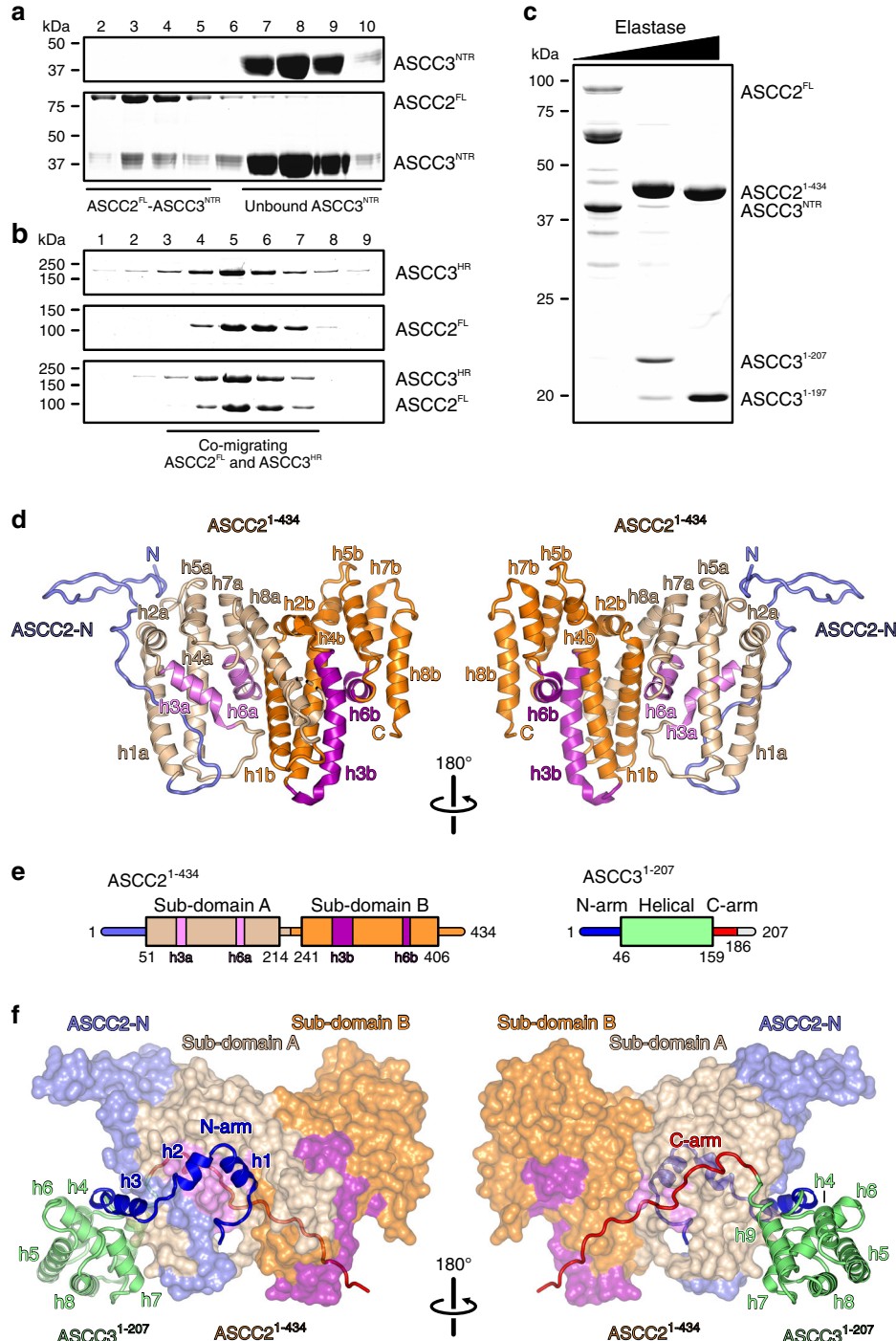

**Fig. 1 Delineation of a minimal ASCC2–ASCC3 complex and structural overview. a, b** SDS-PAGE analysis of SEC runs monitoring interaction of ASCC2$^{FL}$ and ASCC3$^{NTR}$ (**a**) and lack of interaction between ASCC2$^{FL}$ and ASCC3$^{HR}$ (**b**). In this and following figures: kDa, molecular weights of molecular weight markers in kDa. **c** SDS-PAGE analysis of elastase treatment of the ASCC2$^{FL}$-ASCC3$^{NTR}$ complex. Bands running between ASCC2$^{FL}$ and ASCC2$^{1-434}$ represent intermediate ASCC2 fragments that occur transiently in the course of the digestion. Experiments shown in **a–c** have been repeated independently at least three times with similar results. **d** Orthogonal cartoon plots of ASCC2$^{1-434}$. ASCC2-N, N-terminal extension, slate blue; helices in sub-domain A, beige and violet; helices in sub-domain B, orange and purple. Helices are labeled as in the text. N, N-terminus; C, C-terminus. **e** Schemes of the domain organizations of ASCC2$^{1-434}$ and ASCC3$^{1-207}$. Violet and purple bars in the ASCC2$^{1-434}$ scheme represent helices h3a/b and h6a/b. The C-terminal 21 residues of ASCC3$^{1-207}$ (gray line) are not defined in the electron density. **f** Orthogonal views on the ASCC2$^{1-434}$-ASCC3$^{1-207}$ complex, with ASCC2$^{1-434}$ as surface view and ASCC3$^{1-207}$ as cartoon. ASCC2$^{1-434}$, colored as in **d**. N-arm of ASCC3$^{1-207}$, blue; C-arm of ASCC3$^{1-207}$ complex, red; helical domain of ASCC3$^{1-207}$, lime green.

sub-domains are composed of eight α-helices (h1a-h8a/h1b-h8b; Fig. 1d). Within each sub-domain, h1/h2, h4/h5, and h7/h8 form helical hairpins (beige/orange in Fig. 1d) that are connected by h3 and h6 (pink/magenta in Fig. 1d). h3a is significantly shorter than h3b (13 vs. 26 residues, respectively) and connected to h4a by a 13-residue, irregularly structured loop, as opposed to the single-residue connection between h3b and h4b. Furthermore, helices h7a and h8a are pried open compared to the h7b-h8b equivalents, due to helix h1b that wedges in between them (Fig. 1d).

The ASCC3$^{1-207}$ fragment folds into a central helical domain (residues 46–159) and extended arms at the N-termini and C-termini (N-arm, residues 1–45, C-arm, residues 160–186), with which it clasps the compact ASCC2$^{1-434}$ (Fig. 1e, f). The N-arm of ASCC3$^{1-207}$ forms three helices, h1, h2 and h3, preceded and connected by irregularly structured linkers along one flank of ASCC2$^{1-434}$ (Fig. 1f). The first helix (h1, residues 5–13) is embedded between helices h6a and h8a of ASCC2$^{1-434}$ sub-domain A and helix h1b of ASCC2$^{1-434}$ sub-domain B. Helix h2 (residues 15–21) runs across ASCC2$^{1-434}$ helix h3a. Helix h3 (residues 26–40) secures part of the N-terminal extension of ASCC2$^{1-434}$ between helices h1a and h3a of ASCC2$^{1-434}$ sub-domain A (Fig. 1f). The central helical region of ASCC3$^{1-207}$ consists of six helices (h4-h9). Helices h4, h5, h7, h8, and h9 are arranged in a circle around the central helix h6, in front of helix h2a of ASCC2$^{1-434}$ (Fig. 1f). The C-arm stretches in an extended, irregular conformation along the opposite flank of ASCC2$^{1-434}$, connecting sub-domains A and B of ASCC2$^{1-434}$ (Fig. 1f).

**Interfaces between ASCC3 and ASCC2 are evolutionarily conserved.** Structurally or functionally important regions in proteins are often evolutionarily conserved. We analyzed conservation of the ASCC2$^{1-434}$–ASCC3$^{1-207}$ contact regions using the ConSurf server[40]. This analysis revealed that residues located at ASCC2$^{1-434}$–ASCC3$^{1-207}$ interface regions are highly conserved (Fig. 2). Residues 91, 93, 103, 163, and 251 of ASCC2$^{1-434}$ were assigned the highest conservation scores. These residues are located in helix h3a, h6a and h1b, and form a binding pocket for helix h1 of the ASCC3$^{1-207}$ N-arm (Fig. 2a, left). ASCC2$^{1-434}$ residues that form a binding surface for the ASCC3$^{1-207}$ C-arm are conserved to a lesser extent (Fig. 2a, right). Mirroring the conservation pattern on ASCC2$^{1-434}$, residues of the ASCC3$^{1-207}$ N-arm, in particular the first 16 residues, are significantly more conserved compared to other surface positions of the protein (Fig. 2b). This analysis suggests that interactions involving the N-arm of ASCC3$^{1-207}$ may be of particular importance for the formation of the ASCC2$^{1-434}$-ASCC3$^{1-207}$ complex, and that the ASCC2$^{1-434}$–ASCC3$^{1-207}$ interaction observed here is likely conserved in all organisms that contain these proteins.

**The N-terminal arm of ASCC3 is essential for stable binding to ASCC2.** To elucidate the relative importance of different regions of ASCC3$^{1-207}$ for interaction with ASCC2$^{1-434}$, we used structure-informed mutagenesis in combination with analytical SEC and isothermal titration calorimetry (ITC). First, we designed sequential N-terminal truncations of ASCC3$^{1-207}$ with or without the C-arm. ASCC3 fragments lacking C-terminal residues not visible in the structure (ASCC3$^{1-197}$) or additionally lacking the C-arm (ASCC3$^{1-161}$) maintained stable interaction with ASCC2$^{1-434}$ in SEC (Fig. 3a). By contrast, deletion of the N-arm (ASCC3$^{42-197}$) abolished stable binding to ASCC2$^{1-434}$ in SEC (Fig. 3b), while a partial N-arm deletion variant (ASCC3$^{16-197}$) still co-migrated with ASCC2$^{1-434}$ in SEC (Fig. 3c).

Quantifying binding affinities by ITC revealed a similar, yet more detailed, picture. ITC showed a $K_d$ of 3.5 nM for the interaction of the complete ASCC3$^{NTR}$ and ASCC2$^{FL}$ (Fig. 3d). ASCC3$^{1-197}$

(lacking the C-terminal 10 residues of ASCC3$^{1-207}$ but containing all ASCC3 residues with well-defined electron density in the ASCC2$^{1-434}$-ASCC3$^{1-207}$ complex structure) and ASCC2$^{1-434}$ interacted with a similar affinity ($K_d$ of 3.8 nM), suggesting that the fragments contained in our crystal structure encompass the entire ASCC2–ASCC3 interacting regions (Fig. 3e). Deletion of the entire C-arm of ASCC3$^{1-207}$ (ASCC3$^{1-161}$) led to an approximately 14-fold decreased affinity ($K_d = 47.7$ nM; Fig. 3f). Truncation of the N-terminal 15 residues that form helix h1 in the N-arm of ASCC3$^{1-197}$ reduced affinity to ASCC2$^{1-434}$ by more than two orders of magnitude ($K_d = 483.0$ nM; Fig. 3g). Lack of the entire N-arm of ASCC3$^{1-197}$ (ASCC3$^{42-197}$) completely abrogated the interaction, as no signal was detected in ITC measurements (Fig. 3h). Together, the above results indicate that the N-arm, and in particular the first 15 residues, of ASCC3$^{1-207}$ are essential for a stable binary interaction with ASCC2$^{1-434}$, consistent with its high degree of evolutionary conservation. The C-arm of ASCC3$^{1-207}$ contributes to the interaction with ASCC2$^{1-434}$ but it is not essential for the proteins to maintain a stable complex in SEC, consistent with a reduced but still high level of evolutionary conservation of the C-arm.

We next tested whether the interaction pattern observed using recombinant proteins in vitro also applies to the ASCC2–ASCC3 interaction in living cells. Expression constructs encoding C-terminally flag-tagged ASCC3$^{FL}$ and stepwise N-terminally truncated fragments (ASCC3$^{16-end}$, ASCC3$^{42-end}$, ASCC3$^{207-end}$, and ASCC3$^{401-end}$) were used to generate stably transfected HEK293 Flp-In cell lines for the tetracycline-inducible expression of these proteins. After induction, flag-tagged ASCC3 or fragments were captured on anti-flag beads and co-precipitation of ASCC2 was monitored by western blotting. In line with our in vitro interaction mapping, ASCC3 variants with increasing N-terminal deletions co-precipitated stepwise reduced amounts of ASCC2 (Fig. 3i). ASCC3 variants lacking 15, 41, 206, or 400 N-terminal residues, co-precipitated 73, 40, 26, or 16%, respectively, of the amount of ASCC2 associated with full-length ASCC3 (Fig. 3j). While based on our structural and mutagenesis data truncation of the N-terminal 206 or 400 residues of ASCC3 should abrogate direct ASCC2–ASCC3 interactions, ASCC3$^{207-end}$ and ASCC3$^{401-end}$ most likely still pulled down reduced amounts of ASCC2 (Fig. 3i, j), because other ASCC subunits also mediate indirect ASCC2–ASCC3 interactions in vivo.

**ASCC2 and ASCC3 residue substitutions found in human cancers cluster at interfaces and lead to reduced affinities.** 223 and 652 somatic nonsense, missense or frame shift mutations in human cancer cell or tissue samples have been mapped to the *ascc2* and *ascc3* coding regions, respectively (https://cancer.sanger.ac.uk/cosmic). 123 and 95 of these mutations affect residues in ASCC2$^{1-434}$ and ASCC3$^{1-207}$, respectively. Strikingly, 16 of the missense mutations and three nonsense mutations affect residues E53, R58, T60, D65, L70, R96, Y97, D103, or R121 in ASCC2$^{1-434}$, and 17 or one, respectively, of these mutations affect residues R5, R11, S12, D28, R33, K165, or E181 in ASCC3$^{1-207}$, which all constitute direct contact points between the proteins in our crystal structure (Fig. 4a). Many of the remaining point mutations map to, and are expected to disturb, the globular portions of the proteins. In addition, there are three frameshift mutations that would affect almost the entire ASCC2$^{1-434}$ region or large parts thereof (E30, F124, G171 in ASCC2; Fig. 4a). 25 frameshift mutations map to residues F163 and G164 in ASCC3, affecting the entire C-arm of ASCC3$^{1-207}$ (Fig. 4a).

We thus surmised that many residue substitutions of ASCC2$^{1-434}$ or ASCC3$^{1-207}$ found in human cancers affect the

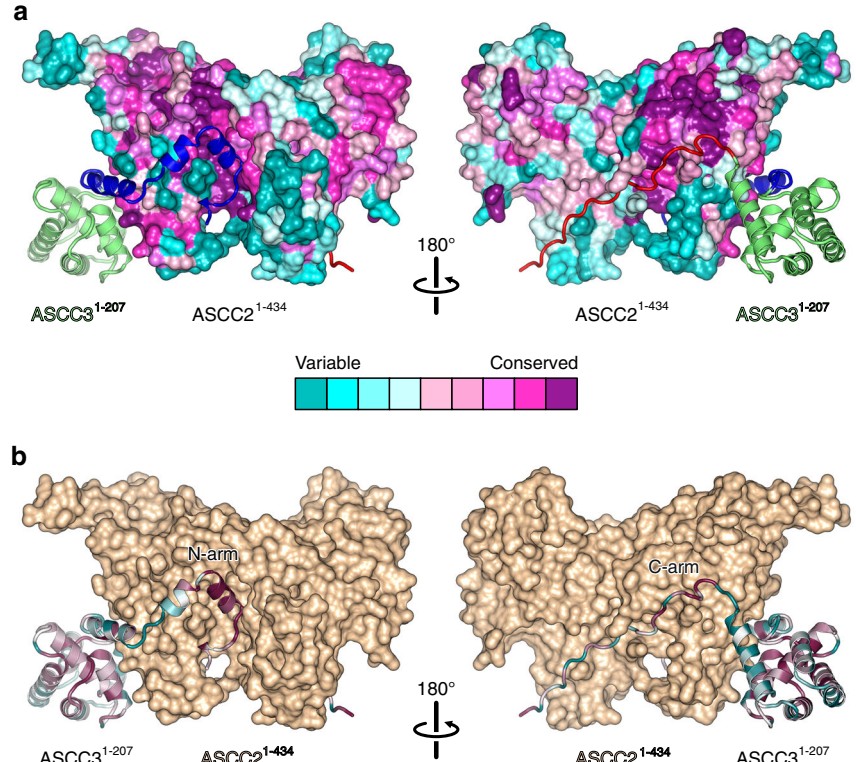

**Fig. 2 Conservation of interfaces. a** Residue conservation mapped to the surface of ASCC2[1–434] with ASCC3[1–207] shown as cartoon (colored as in Fig. 1f). **b** Residue conservation mapped on the structure of ASCC3[1–207] shown as cartoon, with ASCC2[1–434] in surface view (beige). Central legend, color code representing the degree of conservation of individual residues. Views as in Fig. 1d, f.

ASCC2–ASCC3 affinity. To test this hypothesis, we conducted comparative ITC analyses using ASCC2[1–434] and peptides representing the N-terminal 22 residues of ASCC3 exhibiting wild-type (WT) sequence or residue substitutions R5G, R5C, R5H, R5L, R11C, or R11H. R5 residue substitutions are found in large intestine and lung adenocarcinomas, as well as in cervical and esophageal squamous cell carcinomas, while the R11 residue substitutions occur in endometrioid carcinoma and large intestine adenocarcinoma (https://cancer.sanger.ac.uk/cosmic). In our crystal structure, R5 of ASCC3 is positioned at the N-terminus of helix h1 and interacts with D103 of ASCC2[1–434] (D103 of ASCC2[1–434] is also affected by somatic cancer mutations), while R11 of ASCC3 lies at the C-terminal end of helix h1 and engages in hydrogen bonds and/or salt bridges with D63 and D92 in ASCC2[1–434] (Fig. 4b). We used ASCC3 peptides in these experiments as the high affinity observed in ITC between ASCC2[1–434] and ASCC3[1–197] or longer ASCC3 variants (Fig. 3d, e) may mask differences due to single residue substitutions.

WT ASCC3[1–22] bound to ASCC2[1–434] with a $K_d$ of 2.0 μM (Fig. 4c). ASCC3[1–22] peptides bearing R5L or R5G substitutions weakened the interaction with ASCC2[1–434] approximately eight-fold and eleven-fold, respectively (Fig. 4d, e). ASCC3[1–22] peptides comprising R5H or R5C substitutions showed more than 20-fold reduced affinities compared to WT (Fig. 4f, g). R11H or R11C substitutions in ASCC3[1–22] completely eradicated binding to ASCC2[1–434] (Fig. 4h, i). These observations are consistent with the notion that reduced affinity to ASCC2 represents a means by which of R5 and R11 residue substitutions in ASCC3 contribute to cancer phenotypes.

**The N-terminal cassette is an active helicase unit in ASCC3.** ASCC3 bears close resemblance to the spliceosomal RNA helicase, Brr2. Both ASCC3 and Brr2 contain an approximately 400-residue NTR followed by two Ski2-like helicase cassettes with identical domain composition (Fig. 5a, left). It is well documented that the N-terminal Ski2-like helicase cassette constitutes the active helicase unit in Brr2, while the C-terminal cassette lacks ATPase and helicase activities[32]. The reverse situation has been reported for human ASCC3; in contrast to an inactive isolated N-terminal cassette construct, an isolated C-terminal cassette construct was found to be active in DNA duplex unwinding[8]. As we did not perceive alterations in conserved helicase motifs in the N-terminal cassette that would obviously preclude helicase activity (Fig. 6a), we revisited the question of helicase activity of the ASCC3 helicase cassettes.

While we were able to produce a recombinant ASCC3 fragment encompassing the N-terminal cassette (ASCC3[NC]; residues 401–1300), a fragment containing only the C-terminal unit could not be produced in soluble form. To test helicase activities, we pre-incubated ASCC3 variants (see below) with a 12-base pair DNA duplex containing a 31-nucleotide 3'-single-stranded overhang, and bearing a fluorophore (Alexa Fluor 488) on the 3'-terminus of the short strand and a quencher (Atto 540 Q) on the 5'-terminus of the long strand (Fig. 5b). As a control, we first tested the assay with Brr2 variants and an analogous RNA duplex. Fast mixing of Brr2[FL] or Brr2[HR] (residues 395–2129; a truncated version of Brr2 lacking most of the auto-inhibitory NTR, largely equivalent to ASCC3[HR]) pre-incubated with the RNA substrate and ATP in a stopped-flow device led to time-dependent increases in the Alexa Fluor 488 fluorescence, indicating RNA duplex unwinding (Fig. 5c). Unwinding occurred at a significantly higher rate with Brr2[HR] than with Brr2[FL] (Fig. 5c), showing that the stopped-flow/fluorescence setup can reliably monitor differences in helicase activity between alternative dual-cassette helicase constructs. As expected, ASCC3[HR]

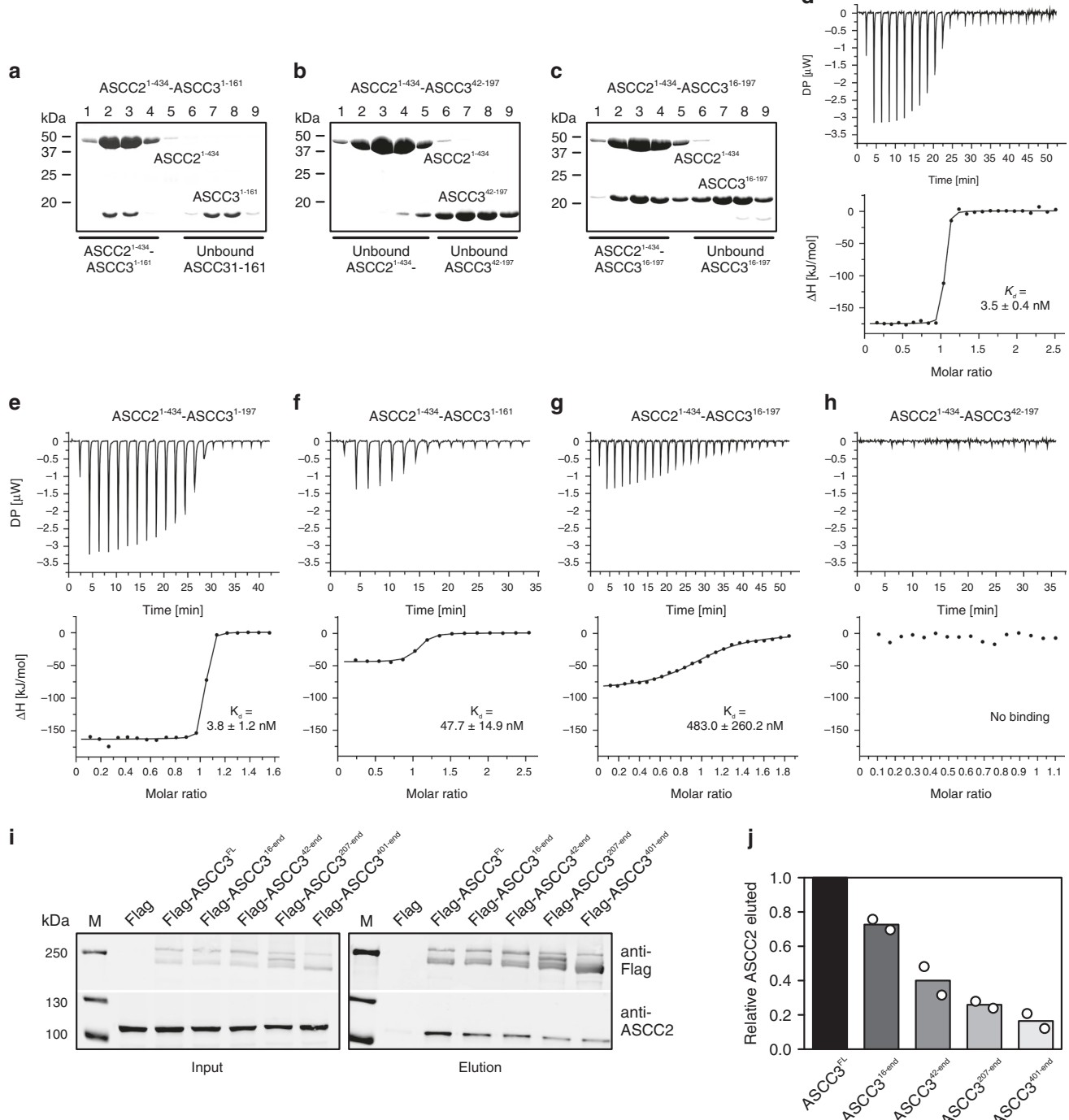

**Fig. 3 Interaction analysis. a–c** SDS-PAGE analysis monitoring SEC runs of the indicated mixtures of proteins. **d–h** ITC runs monitoring the interaction between the indicated pairs of proteins. Deduced $K_d$ values are listed as means ± SD for runs for which affinities could be quantified. **i** Western blot monitoring pulldown of ASCC2 by the indicated ASCC3 constructs from cell extracts. **j** Quantification of the data shown in **i**. Columns represent means relative to ASCC3[FL] of $n = 2$ independent experiments, using the same biochemical samples. Open circles, individual measurements.

efficiently unwound a DNA duplex with 3'-single stranded overhang, but not a duplex with 5'-single stranded overhang (Fig. 5d), and preferentially used ATP for DNA unwinding (Fig. 5e). Surprisingly, however, in our hands, ASCC3[NC] also exhibited helicase activity, albeit reduced compared to ASCC3[HR] (Fig. 5f). These findings were confirmed using gel-based unwinding assays (Supplementary Fig. 2).

Nucleic acid helicases contain a number of conserved, functionally important sequence motifs in their RecA domain

cores (Fig. 6a). In particular, a lysine in motif I is required for ATP binding, while an aspartate in motif II is crucial for coordinating a magnesium ion to trigger the hydrolysis of ATP (Fig. 6a), and substitutions of these residues are expected to strongly interfere with helicase activity. To further test the contributions of the N-terminal and C-terminal cassettes to the helicase activity of a dual-cassette ASCC3 construct, we individually substituted the corresponding lysine and aspartate residues in the N-terminal cassette (K505, D611), in the

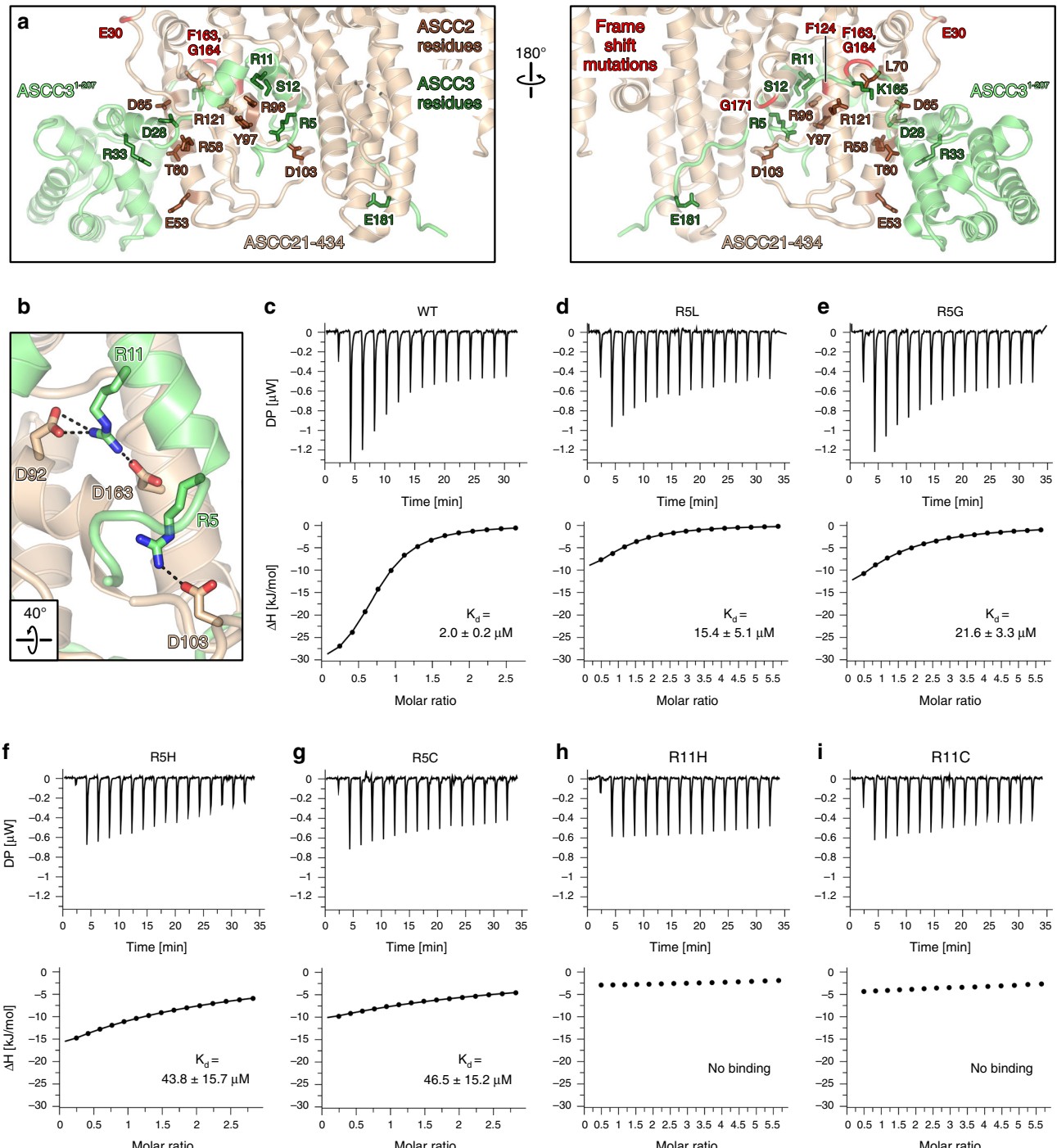

**Fig. 4 Interface residues in ASCC2 and ASCC3 affected by somatic cancer mutations. a** Orthogonal views on the ASCC2$^{1-434}$-ASCC3$^{1-207}$ structure, highlighting interface residues and frameshift positions affected by somatic cancer mutations. ASCC2$^{1-434}$, beige; ASCC3$^{1-207}$, lime green; ASCC2 residues, brown sticks; ASCC3 residues, dark green sticks; first positions affected by frameshift mutations, red. Orientations of the panels as in Fig. 1d,f. **b** Details of the interaction networks involving cancer-related residues D103 of ASCC2, as well as R5 and R11 of ASCC3. Dashed lines, salt bridges. Rotation symbol, view relative to Fig. 1d,f, left. **c-i** ITC runs monitoring the interaction of ASCC2$^{1-434}$ with the indicated ASCC3$^{1-22}$ peptide variants. Deduced $K_d$ values are listed as means ± SD for runs for which affinities could be quantified.

C-terminal cassette (K1355, D1453) or in both cassettes of ASCC3$^{HR}$ with alanine or asparagine, respectively. We verified the intended mutations by sequencing the final generation of baculoviruses used for production of the ASCC3$^{HR}$ variants (Supplementary Fig. 3). ASCC3$^{HR}$ variants that contained mutations in motifs I or II of the N-terminal cassette had strongly reduced or no detectable helicase activities (Fig. 6b). By

contrast, ASCC3$^{HR}$ variants that contained residue substitutions K1355N or D1453A within the C-terminal cassette retained the activity of WT ASCC3$^{HR}$ (Fig. 6c). Again, gel-based unwinding assays using D611A and D1453A variants of ASCC3$^{HR}$ were consistent with these results (Supplementary Fig. 2). Also consistent with these findings, DNA helicase activities of the K505N/K1355N and D611A/D1453A double variants were

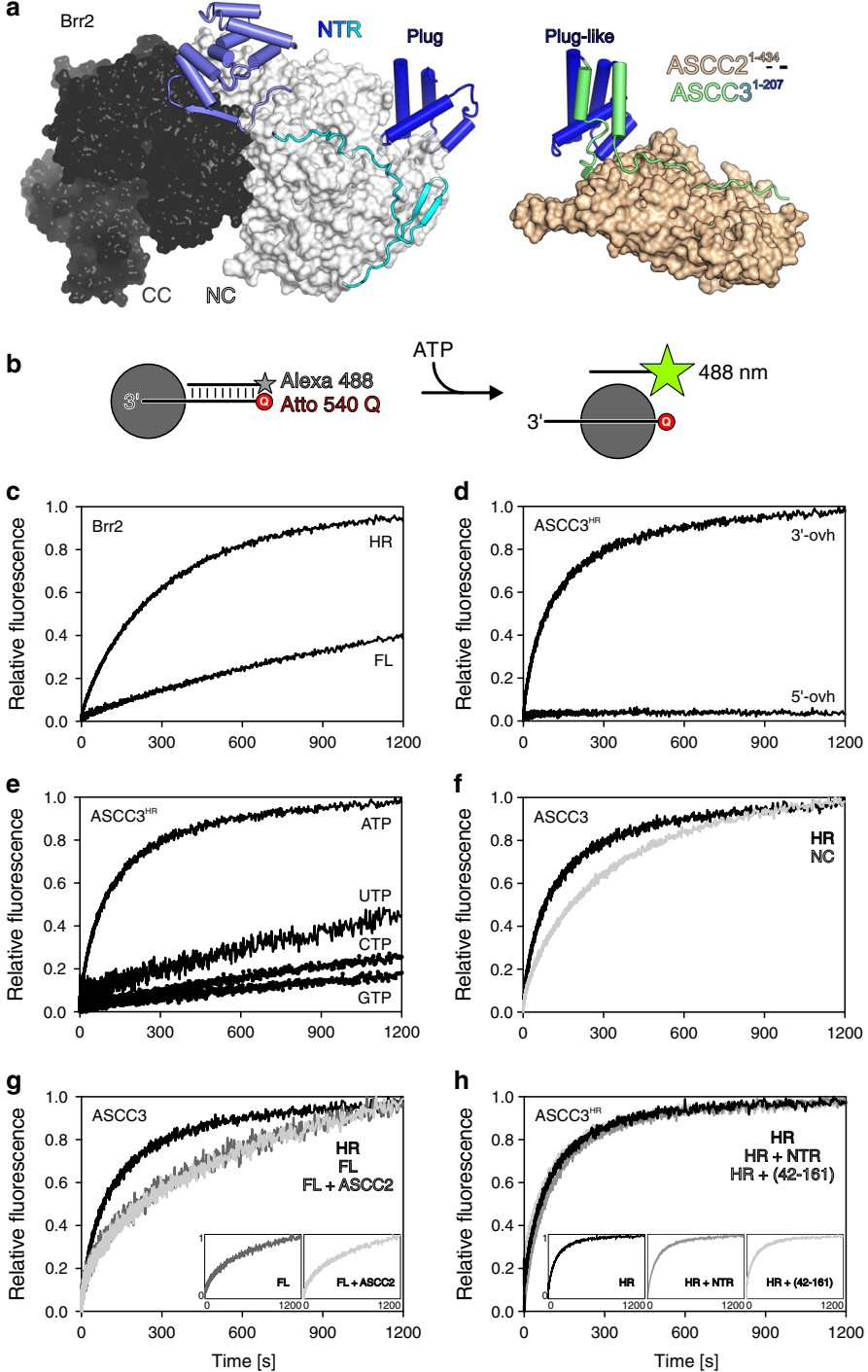

**Fig. 5 Unwinding assays. a** Comparison of the structure of full-length yeast Brr2 (left) and the present ASCC2$^{1-434}$-ASCC3$^{1-207}$ complex (right) after superposition of the plug/plug-like domains (blue). Human Brr2 in the U4/U6•U5 tri-snRNP or in the pre-catalytic spliceosome exhibits a plug domain very similar to yeast Brr2 blocking RNA access[42,43] (PDB IDs 3JCR, 6QW6, 6QX9). NC, N-terminal cassette; CC, C-terminal helicase cassette. **b** Experimental setup for stopped-flow/fluorescence-based unwinding assays. Gray sphere, helicase; star symbol, fluorophore (Alexa 488); Q, quencher (Atto 540 Q). **c** Stopped-flow/fluorescence-based assays monitoring unwinding of a 3′-overhang RNA by Brr2$^{FL}$ or Brr2$^{HR}$, showing that auto-inhibition, which has been documented for Brr2 using gel-based unwinding assays[33], can be readily detected using the present experimental setup. **d** Stopped-flow/ fluorescence-based assays monitoring unwinding of DNA bearing a 3′-overhang (3′-ovh), and lack of unwinding of a 5′-overhang (5′-ovh) DNA by ASCC3$^{HR}$ upon addition of ATP. **e** Nucleotide preference of ASCC3$^{HR}$ in unwinding a 3′-overhang DNA. **f** Stopped-flow/fluorescence-based assays monitoring unwinding of 3′-overhang DNA by ASCC3$^{NC}$ compared to ASCC3$^{HR}$ using ATP. **g** Stopped-flow/fluorescence-based assays monitoring unwinding of a 3′-overhang DNA by the indicated ASCC3 constructs in the absence or presence of ASCC2 using ATP. Insets, side-by-side presentation of the largely overlapping curves. **h** Stopped-flow/fluorescence-based assays monitoring unwinding of a 3′-overhang DNA by ASCC3$^{HR}$ alone or in the presence of ASCC3$^{NTR}$ or ASCC3$^{42-161}$ using ATP. Insets, side-by-side presentation of the largely overlapping curves. FL/HR/NC/NTR/(42-161), ASCC3 variants as defined in the text.

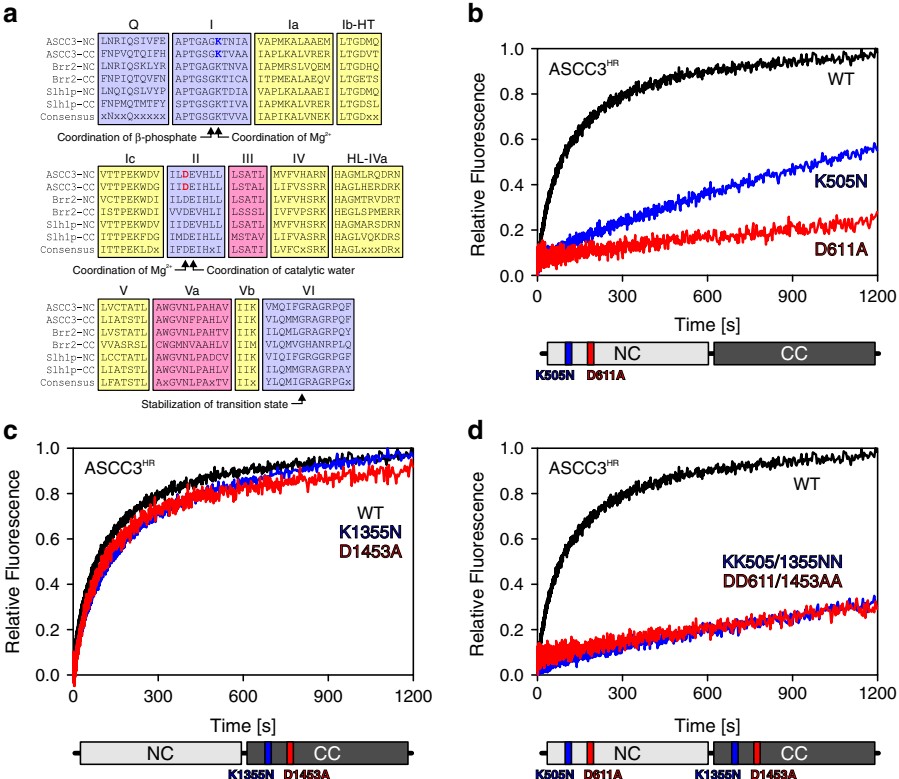

**Fig. 6 Mutational analysis. a** Multiple sequence alignment of conserved helicase motifs (indicated by letters or Roman numerals above the alignment) in human ASCC3, human Brr2 and yeast Slh1p N-terminal cassettes (NC) and C-terminal cassettes (CC). The conserved motif I lysine and motif II aspartate residues of ASCC3 cassettes, which were altered in the present analysis, are highlighted in blue and red, respectively. Color coding of the motifs: involved in ATP binding, blue; involved in RNA binding, yellow; involved in communication of RNA binding and ATPase activities, red. Roles of selected residues are indicated below the alignment. **b–d** Stopped-flow/fluorescence-based assays monitoring unwinding of 3′-overhang DNA by the indicated ASCC3$^{HR}$ variants using ATP. Schemes below the unwinding traces represent ASCC3$^{HR}$ with altered residues highlighted. WT/K505N/D611A/K1355N/D1453A/ KK505/1355NN/DD611/1453AA, ASCC3$^{HR}$ variants as defined in the text.

almost fully abrogated (Fig. 6d). These results suggest that, contrary to previous observations[8] and similar to Brr2 (ref. [32]), the N-terminal cassette of ASCC3 is an active helicase in vitro. As the ASCC3$^{NC}$ helicase activity was reduced compared to that of ASCC3$^{HR}$ (Fig. 5f; Supplementary Fig. 2) it remains to be seen whether the C-terminal cassette is inactive but stimulates helicase activity of ASCC3$^{NC}$, as has been observed for Brr2 (ref. [32]), or whether ASCC3$^{CC}$ is an active helicase as well that contributes to the overall activity of ASCC3$^{HR}$.

**ASCC2 does not influence the helicase activity of ASCC3.** The Brr2 NTR contains helical "plug" and PWI-like domains connected by extended, intrinsically disordered regions[33,41]. In isolated yeast Brr2 (ref. [33]), as well as in human Brr2 in the U4/U6•U5 tri-snRNP or in the pre-catalytic spliceosome[42,43], the NTR or part of it can fold back onto the helicase region, with the plug domain blocking substrate RNA loading (Fig. 5a, left), thereby auto-inhibiting Brr2 ATPase and helicase activities. Having established a similar functional organization of the helicase regions of ASCC3 and Brr2, we sought to investigate whether ASCC3 is also auto-regulated via its NTR.

While the overall sequence identity between human ASCC3 and human Brr2 is around 41%, the sequence identity is lower for their NTRs, around 20%. Despite the lower level of sequence conservation, structural comparison of ASCC3$^{1-207}$ with the NTR of yeast and human Brr2 revealed that a central, four-helix portion of ASCC3$^{1-207}$ (residue 59–142) closely resembles the Brr2 plug domain (root-mean-square deviation [rmsd] of 2.7 Å for 36 pairs of common Cα atoms; Fig. 5a). However, in our

crystal structure, the ASCC3$^{1-207}$ arms neighboring the plug-like domain extend along ASCC2$^{1-434}$ (Fig. 5a, left); thus, when bound to ASCC2, these regions would not be available to fold back on the ASCC3 helicase region.

The helicase activity of ASCC3$^{FL}$ was significantly reduced compared to ASCC3$^{HR}$, consistent with similar auto-inhibition as for Brr2 (Fig. 5g). When added in trans, ASCC3$^{NTR}$ or the plug domain-containing fragment, ASCC3$^{42-161}$, did not alter ASCC3$^{HR}$ helicase activity (Fig. 5h), indicating that the NTR has to be covalently connected to the helicase region to elicit the auto-inhibitory effect. Addition of ASCC2 did not influence the helicase activity of ASCC3$^{FL}$ (Fig. 5g). Due to the high affinities quantified for the ASCC2$^{FL}$-ASCC3$^{NTR}$ and ASCC2$^{1-434}$-ASCC3$^{1-197}$ interactions (Fig. 3d, e), we consider it unlikely that the NTR is sequestered from ASCC2 by interacting in a mutually exclusive manner with the helicase region. Instead, the plug-like domain in ASCC3 may still be able to occupy an inhibitory position when associated with ASCC2.

## Discussion

In this study, we have delineated interacting regions of the ASCC2 and ASCC3 subunits of the ASCC DNA repair machinery, and elucidated the structural basis for this interaction. Guided by our structural results, we identified regions and residues of the two proteins that significantly contribute to stable complex formation. The importance of regions for the interaction scales with the degree of their evolutionary conservation. Moreover, we observed that a large number of ASCC2 and ASCC3 residue substitutions encoded by somatic mutations in cancers map to

ASCC2–ASCC3 interface regions and that selected, cancer-related residue substitutions in ASCC3 lead to reduced ASCC2–ASCC3 affinity. We also conducted in vitro enzymatic analyses to better characterize the ASCC3 nucleic acid helicase, showing (i) that the N-terminal cassette represents an active helicase, (ii) that helicase activity in the C-terminal cassette is dispensable for the activity of the full-length protein, (iii) that helicase activity in ASCC3 is auto-inhibited by the NTR, and (iv) that interaction with ASCC2 does not strongly influence ASCC3 DNA unwinding activity in vitro.

ASCC3 and Brr2 both belong to a sub-group of the Ski2-like family of SF2 helicases with a tandem array of complex helicase units. The helicase regions of ASCC3 and Brr2 seem to be organized in an analogous fashion, comprising an active N-terminal and an inactive, or at least largely dispensable, C-terminal helicase cassette. Consistent with this notion, an ASCC3 variant bearing an ATPase-disrupting substitution in the C-terminal helicase cassette, but not in the N-terminal cassette, can rescue a ribosome poly-A readthrough phenotype elicited by ASCC3 knockdown[25]. Likewise, in the yeast ASCC3 homolog, Slh1p, residue substitutions in the N-terminal cassette can abrogate ribosome quality control function in vivo[27].

Several proteins have been identified that directly bind to the helicase region of Brr2 and modulate its helicase activity, including a C-terminal Jab1-like domain of the Prp8 protein[36] and a largely intrinsically unstructured protein, FBP21 (ref. [38]). ASCC1 interacts with the helicase region of ASCC3 but not with the NTR[10]. It will be interesting to see in future studies, which of the cassettes it interacts with, whether it modulates the ASCC3 helicase upon binding and whether similar molecular principles as in the case of Brr2 and its protein co-factors apply to such putative regulation of ASCC3.

Our structural analysis revealed that ASCC3 and Brr2 share at least some structural organization in their NTRs as well. Both contain a plug/plug-like domain close to their N-termini (residues 59–142 in human ASCC3, 113–192 in yeast Brr2, 107–180 in human Brr2). In Brr2, the plug domain contributes to auto-inhibition of the isolated enzyme[33] and our results suggest similar auto-inhibition in isolated ASCC3. Based on our crystal structure, regions neighboring the ASCC3 plug-like domain are expected to be guided away from the ASCC3 helicase region when bound to ASCC2, however, ASCC2 did not alleviate ASCC3 auto-inhibition. ASCC3 auto-inhibition may, therefore, be predominantly mediated via the plug-like domain, which remains accessible in complex with ASCC2, occupying a position on the helicase region where it competes with DNA substrate loading. Still, our data suggest that a covalent connection of the plug-like domain to the helicase region is required for auto-inhibition. Thus, the general principle of auto-inhibition via the NTR seems to be conserved between ASCC3 and Brr2, but details of the mechanisms may differ.

Correlation of the evolutionary conservation of ASCC2–ASCC3 interfaces and the contributions of these regions to a stable interaction underscore the functional importance of the complex we structurally analyzed. As the two proteins seem to participate together in various ASCC functions (transcription co-activation, DNA repair, ribosome-dependent protein quality control, viral defense), we expect that all of these functions depend on the observed ASCC2–ASCC3 interaction.

We observed an extraordinary number of residues affected by somatic cancer mutations at ASCC2–ASCC3 interfaces, and found that some cancer-related residue substitutions lead to reduced ASCC2–ASCC3 affinity. Based on these observations and previous insights into the role of ASCC components in DNA alkylation damage repair, we suggest that reduced ASCC2–ASCC3 affinity might contribute to malignant transformation. As DNA alkylation damage can occur non-enzymatically[44,45], the increased metabolic activity of cancer cells may lead to a higher level of metabolic DNA damage[46], rendering cancer cell proliferation dependent on systems that can repair excessive DNA damage. Indeed, ASCC3 is over-expressed in several cancers and ASCC3 knockdown, which would be expected to lead to less efficient AlkBH3-mediated DNA de-alkylation repair, has been shown to negatively impact tumor cell proliferation in culture and xenograft models[8,13]. Other components of ASCC seem to contribute to efficient ASCC3/AlkBH3-mediated DNA alkylation damage repair by establishing a complex DNA damage signaling and repair pathway[9,10]. Specifically, ASCC1 appears to promote co-localization of ASCC2 and ASCC3 at nuclear foci during alkylation damage, and loss of ASCC1 leads to increased alkylation sensitivity[10]. Proper co-localization of ASCC2 and ASCC3, therefore, appears to be essential for efficient DNA alkylation damage repair. Similar to ASCC1 knockdown, reduced ASCC2–ASCC3 affinity, as elicited by several somatic cancer mutations, may influence ASCC2–ASCC3 co-localization at nuclear foci, leading to increased DNA alkylation damage. This model does not invoke a direct influence of ASCC2 on ASCC3 helicase activity, consistent with our findings. While reduced DNA alkylation damage repair due to reduced ASCC2–ASCC3 affinity would be expected to negatively impact cancer cell proliferation, DNA alkylation can represent mutagenic lesions[44,45]. Thus, reduced ASCC2–ASCC3 interaction may contribute to the initial development of a cancer cell phenotype, as increased DNA alkylation damage may lead to transforming mutations.

## Methods

**Molecular cloning.** DNA regions encoding the proteins/protein regions of interest were PCR-amplified from synthetic genes (GeneArt), optimized for expression in *Trichoplusia ni* cells. Supplementary Tables 2 and 3 list sequences of PCR primers and synthetic genes, respectively. DNA regions encoding ASCC3[FL], ASCC3[NTR], ASCC3[HR], ASCC3[NC], ASCC3[1–207], or full-length ASCC2 were cloned into a modified pFL vector under control of the very late polyhedrin promoter, which directed the synthesis of fusion proteins with TEV-cleavable N-terminal His$_{10}$-tags. A DNA fragment encoding ASCC2[1–434] was cloned into the pUCDM vector and Cre-recombined with ASCC3[1–207]-encoding pFL. The QuikChange II XL Site-Directed Mutagenesis Kit (Agilent) was used to introduce mutations that give rise to ASCC3[HR] variants K505N, K1355N, D611A, D1453A, K505N-K1355N, and D611A-D1453A. The expression cassettes were integrated into the MultiBac baculoviral genome via Tn7 transposition within a lacZα gene, allowing selection of recombinants by blue/white screening. Recombinant bacterial artificial chromosomes (BACs) were isolated from the bacterial hosts.

DNA fragments encoding ASCC3[NTR], ASCC3[1–207], ASCC3[1–197], ASCC3[1–161], ASCC3[16–197], ASCC3[42–197], and ASCC2[1–434] were PCR-amplified from the pFL vectors encoding ASCC3[NTR] or full-length ASCC2. These fragments were inserted into the pETM-11 vector (EMBL, Heidelberg) that guides production of proteins with TEV-cleavable N-terminal His$_6$-tags. All constructs were confirmed by sequencing.

**Protein production and purification.** ASCC2[1–434] was co-produced with ASCC3[1–207] in High Five cells. For virus production, the isolated BAC DNA was transfected into Sf9 cells in a six-well plate. After 68 h, the supernatant containing the initial virus ($V_0$) was collected and used to infect a 50 ml suspension culture of Sf9 cells to amplify the virus ($V_1$). For large scale production, 400 ml of High Five cell culture in a 2 l shaker flask was infected with 1 ml of $V_1$ virus, and cells were harvested after 3–4 days. Production of the intended ASCC3[HR] variants was confirmed by sequencing of PCR-amplified regions from the $V_1$ baculoviruses.

For production of isolated ASCC2[1–434] and of isolated N-terminal ASCC3 fragments, *E. coli* BL21 (DE3) cells were transformed with the corresponding pETM-11 vectors, cultivated in auto-inducing medium[47] at 18 °C and harvested at an OD$_{600}$ of ~10. SeMet-labeled ASCC2[1–434] protein was produced by auto-induction in cells grown in PASM-5052 medium[48], which contained 10 µg/mg of unlabeled methionine and 125 µg/ml of SeMet. Cells were harvested by centrifugation and stored at −20 °C.

The same purification protocol was used for all individual proteins of interest (POIs) and for the ASCC2[1–434]-ASCC3[1–207] used for crystallization, unless otherwise specified. Cell pellets were re-suspended in lysis buffer (20 mM HEPES-NaOH, pH 7.5, 500 mM NaCl, 10 mM imidazole, 1 mM DTT, 8.6% [v/v] glycerol), supplemented with protease inhibitors and lysed by sonication using a Sonopuls Ultrasonic Homogenizer HD (Bandelin). After centrifugation and filtration, the soluble fraction was incubated with Ni$^{2+}$-NTA beads for 1 h at 4 °C, and loaded into a gravity flow column. After extensive washing of the beads with lysis buffer,

the POI was eluted using lysis buffer containing 400 mM imidazole. The $His_{6/10}$-tags of the POI was cleaved off during overnight dialysis into lysis buffer without imidazole at 4 °C. The protease-treated sample was reloaded on a fresh $Ni^{2+}$-NTA gravity flow column to remove the $His_{6/10}$-tag, uncleaved protein and the His-tagged protease. For crystallization, the flow-through was further purified by SEC on a Superdex 200 10/600 GL column (GE Healthcare) in 20 mM HEPES-NaOH, pH 7.5, 100 mM NaCl, 1 mM DTT. For other studies, the flow-through was further purified by SEC on a Superdex 200 10/600 GL column in 20 mM HEPES-NaOH, pH 7.5, 250 mM NaCl, 1 mM DTT, 5% (v/v) glycerol. For $ASCC3^{NTR}$, the SEC step was omitted.

SeMet-labeled $ASCC2^{1-434}$–$ASCC3^{1-207}$ complex was assembled from individually purified, SeMet-labeled $ASCC2^{1-434}$ and unlabeled $ASCC3^{1-207}$ by incubation on ice for 30 min, followed by SEC on a Superdex 200 10/600 GL size-exclusion column (GE Healthcare). $ASCC2^{FL}$–$ASCC3^{NTR}$ complex was formed and purified in the same way.

**Limited proteolysis**. 5.7 μg of purified $ASCC2^{FL}$-$ASCC3^{NTR}$ complex were incubated with increasing amounts (0.0013, 0.013, and 0.13 μg) of different proteases in 20 mM HEPES-NaOH, pH 7.5, 250 mM NaCl, 1 mM DTT, 5% (v/v) glycerol at 4 °C overnight. The reactions were stopped by adding SDS-PAGE loading buffer, and samples were separated by SDS-PAGE. Bands of interest were analyzed by in-gel trypsin digestion and mass spectrometric fingerprinting.

**Crystallographic procedures**. Unmodified or SeMet-modified $ASCC2^{1-434}$-$ASCC3^{1-207}$ complex was crystallized by the hanging drop vapor diffusion technique (0.5 μl protein at 5.8 mg/ml plus 1 μl reservoir) at 4 °C, using a reservoir that contained 0.1 M MES-NaOH, pH 6.5, 15% (w/v) polyethylene glycol 3350, 7% (v/v) 2-methyl-2,4-pentanediol, 3% (v/v) methanol. Crystals were cryo-protected by transfer into mother liquor containing 20% (v/v) propylene glycol, and flash-cooled in liquid nitrogen. Diffraction data were collected at beamline 14.1 of the BESSY II storage ring (Helmholtz-Zentrum Berlin, Germany) at 100 K. Diffraction data were processed with XDS[49]. The structure of $ASCC2^{1-434}$-$ASCC3^{1-207}$ complex was solved by the SeMet SAD strategy using phenix.autosol[50]. An initial model was calculated using phenix.autobuild[51]. The model was completed through alternating rounds of manual model building using Coot[52] and automated refinement using phenix.refine[53]. Data collection, structure solution and refinement statistics are listed in Supplementary Table 1. Structure figures were prepared using PyMOL (Version 1.8 Schrödinger, LLC).

**Analytical size exclusion chromatography**. Proteins were produced and purified individually. 1 μmol $ASCC2^{1-434}$ was mixed with 2 μmol of $ASCC3^{1-161}$, $ASCC3^{42-197}$, or $ASCC3^{16-197}$ in 20 mM HEPES-NaOH, pH 7.5, 100 mM NaCl, 1 mM DTT to final volume of 50 μl. The mixtures were incubated on ice for 30 min and then chromatographed on a Superdex 200 3.2/300 analytical size exclusion column (GE Healthcare). Fifty microliter fractions were collected and subjected to SDS-PAGE analysis.

**Isothermal titration calorimetry**. ITC analyses were conducted on an iTC200 instrument (MicroCal). Proteins were dialyzed against ITC buffer (20 mM HEPES-NaOH, pH 7.5, 100 mM NaCl). 300 μl of ASCC3 variant solution (10 μM according to UV absorption) were loaded into the sample cell, and were titrated with ASCC2 variant solution (100 μM) in the syringe via 18 injections of 1.5 μl each. Peptides representing the N-terminal 22 residues of ASCC3 (WT and R5H, R5C, R5G, R5L, R11H, R11C variants) were chemically synthesized (GL biochem) with C-terminal amination. Peptides were dissolved in ITC buffer. 300 μl peptide solution (10–15 μM according to weighing) were loaded into the sample cell, and were titrated with $ASCC2^{1-434}$ solution (200 μM or 400 μM) in the syringe via 16 injections of 2.5 μl each. All measurements were conducted at 25 °C. Data were analyzed with MicroCal PEAQ-ITC Analysis Software. For data analysis, stoichiometries were fixed to 1, as known from our crystal structure analysis, and the concentrations of ASCC3 variants or peptides were allowed to vary.

**Generation of HEK293 Flp-In™ T-REx™ cell lines and immunoprecipitation of complexes**. pcDNA5-based constructs for the expression of C-terminally $His_6$-PreScission protease cleavage site-2xFlag (Flag)-tagged full-length or N-terminally truncated ASCC3 variants were transfected into HEK293 Flp-In™ T-REx™ cells (Invitrogen) according to the manufacturer's instructions. Cells in which the transgene had been genomically integrated into a specific locus were selected using hygromycin and blasticidin, and population cell lines were generated. Cells were grown in Dulbecco's modified Eagle Medium supplemented with 10% fetal bovine serum at 37 °C with 5% $CO_2$, and expression of the tagged proteins was induced by addition of 1 μg/ml tetracycline for 24 h before harvesting.

For immunoprecipitation of complexes containing Flag-tagged proteins[54–56], cells expressing full-length or truncated ASCC3-Flag or the Flag tag alone were lysed by sonication in 50 mM TRIS-HCl, pH 7.4, 150 mM NaCl, 0.5 mM EDTA, 0.1% (v/v) Triton-X-100, 10% (v/v) glycerol and cOmplete™ protease inhibitors (Roche). Cell debris were pelleted by centrifugation and the cleared lysate was incubated with anti-FlagM2 magnetic beads (Sigma-Aldrich) for 2 h. After thorough washing steps, complexes were eluted using 3xFlag peptide. Proteins were

precipitated using 20% (w/v) trichloroacetic acid before separation by SDS-PAGE. Western blotting was performed using antibodies against ASCC2 (Proteintech, 11529-1-AP; diluted 1:1000) and the Flag tag (Sigma-Aldrich, F3165; diluted 1:7500). ASCC2 and ASCC3 bands were quantified using Image J software. The amount of ASCC2 relative to each ASCC3 fragment was calculated and normalized to the relative amount of ASCC2 co-precipitated with $ASCC3^{FL}$ (set at 1).

**Stopped-flow/fluorescence-based helicase assays**. DNA or RNA unwinding was monitored by fluorescence stopped-flow on a SX-20MV spectrometer (Applied photophysics). The DNA and RNA substrates contained a 12-base pair duplex region and a 31-nucleotide 3'-overhang (for sequences see Supplementary Table 4). They harbored an Alexa Fluor 488 moiety on the 3'-end of the short strand and an Atto 540 Q quencher on the 5'-end of the complementary strand, which were in close proximity after annealing. For negative control, we used a 5'-overhang DNA duplex with reverse sequences and labels (Supplementary Table 4). Experiments were conducted at 30 °C in 40 mM TRIS-HCl, pH 7.5, 80 mM NaCl, 0.5 mM $MgCl_2$. 250 nM protein were pre-incubated with 50 nM DNA or RNA duplex for 5 min at 30 °C. 60 μl of the protein–DNA or protein–RNA mixture were rapidly mixed with 60 μl of 4 mM ATP/$MgCl_2$, and the fluorescence signal was monitored for 20 min using a 495 nm cutoff filter (KV 495, Schott). Alexa Fluor 488 was excited at 465 nm. An increase of fluorescence was observed when the duplex was separated. Control experiments included 60 μl of protein–DNA or protein–RNA mixture mixed with buffer, and DNA or RNA duplex alone mixed with ATP/$MgCl_2$. Data were analyzed and plotted using Prism (GraphPad).

**Gel-based helicase assays**. The same 3'-overhang DNA as for stopped-flow/fluorescence-based unwinding assays, but lacking fluorophore and quencher moieties, was used for gel-based unwinding assays (for sequences see Supplementary Table 4). The long strand was 5'-[$^{32}$P] labeled by incubating 5 μl of 20 μM oligonucleotide, 20 μl of γ[$^{32}$P]-ATP (Hartmann Analytic), 3 μl 10× reaction buffer (New England Biolabs) and 2 μl T4 polynucleotide kinase (New England Biolabs) for 1 h at 37 °C. The labeled oligonucleotide was further passed through a Microspin G25 column (Sigma), PCI-extracted and annealed to the short, unlabeled complementary oligonucleotide in a 1:1 molar ratio in annealing buffer (10 mM TRIS-HCl, pH 8.5). Unwinding reactions were carried out at 30 °C. 150 nM purified recombinant ASCC3 variants were mixed with 12 nM DNA substrate in 40 mM TRIS-HCl, pH 7.5, 0.5 mM $MgCl_2$, 187 mM NaCl, 3.75% glycerol, 0.75 mM DTT. After a 10 min incubation at 30 °C, reactions were started by addition of 5 mM ATP/$MgCl_2$. Aliquots were withdrawn at selected time points and reactions were quenched with 50 mM TRIS-HCl, pH 7.5, 50 mM EDTA, 0.5% [w/v] SDS). For 0 min time points, all reactants except ATP/$MgCl_2$ were added. Samples were separated on 12% non-denaturing polyacrylamide gels, DNA bands were visualized using a phosphorimager and quantified with ImageQuant software (Cytiva). The fraction of unwound DNA in each sample was calculated as $I^{ss}/(I^{ss} + I^{ds})$, in which $I^{ss}$ is the intensity of the single-stranded DNA band and $I^{ds}$ is the intensity of the double-stranded DNA band. The fraction unwound at 0 min was subtracted from the fraction unwound for each sample.

**Reporting summary**. Further information on research design is available in the Nature Research Life Sciences Reporting Summary linked to this article.

## Data availability

Structure factors and coordinates have been deposited in the RCSB Protein Data Bank (https://www.rcsb.org/) with accession code 6YXQ. Source data for Figs. 1a–c, 3a–c, i, j, 5c–h, 6 and Supplementary Fig. 2 are provided. Other data are available from the corresponding author upon reasonable request. Source data are provided with this paper.

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

## Acknowledgements

We acknowledge access to beamline BL14.1 of the BESSY II storage ring (Berlin, Germany) via the Joint Berlin MX Laboratory sponsored by Helmholtz Zentrum Berlin für Materialien und Energie, Freie Universität Berlin, Humboldt-Universität zu Berlin, Max-Delbrück Centrum, Leibniz-Institut für Molekulare Pharmakologie and Charité–Universitätsmedizin Berlin. We thank the Biological Mass Spectrometry Unit (Dr. Christoph Weise) at the DFG-funded core facility BioSupraMol of Freie Universität Berlin for mass spectrometric fingerprinting. J.J. was sponsored by a fellowship from the Chinese Scholarship Council. A.J.P.B. was supported by a postdoctoral fellowship of the Alexander von Humboldt Foundation. M.C.W. acknowledges funding by the QBI-FUB Collaborative Integrative Structural Biology Initiative. M.T.B. and K.E.B. acknowledge funding by the Deutsche Forschungsgemeinschaft (BO3442/1-2 to M.T.B. and SFB860 to K.E.B.) and the University Medical Centre Göttingen.

## Author contributions

J.J. performed experiments with help by E.A., N.H., and A.J.P.B., except generation of stable cell lines, pulldown experiments and western blotting, which were performed by P.H. and K.E.B. J.J. and M.C.W. wrote the manuscript with input from the other authors. All authors participated in data interpretation. M.T.B. and M.C.W. coordinated the studies. M.T.B., K.E.B., and M.C.W. provided funding for the studies. Correspondence and requests for materials should be addressed to M.C.W.

## Funding

## Competing interests

The authors declare no competing interests.
