## [Peer Review File · Nature Communications]

REVIEWER COMMENTS

Reviewer #1 (Remarks to the Author):

Jia et al report the interactions of DNA repair factors ASCC2 and ASCC3 through structural, ITC and mutagenesis. They determined a crystal structure of the ASCC21-434-ASCC31-207 complex was determined via SAD approach at 2.7 Å resolution to R_{work}/R_{free} values of 20.4/24.7 %. Their structural analysis suggested the Interfaces between ASCC3 and ASCC2 are evolutionarily conserved. In addition, they demonstrated that the N-terminal arm of ASCC3 is essential for stable binding to ASCC2 by mutagenesis and ITC. Finally, they found that some key amino acid at the interface between ASCC2 and ASCC3 are mutated in human cancers. Overall the structure is good and the interaction data are solid.

However, there are two serious concerns:

1. The authors found that ASCC2 does not influence the helicase activity of ASCC3. If so, what is the connection between the mutations at the interface and cancers?
2. The authors found the NC of ASCC3 has the 3'-5' helicase activity in contrast with the published results that the CC of ASCC3 has the same activity. Could ASCC2 influences the CC activity instead? Can the authors explain why they got conflicting helicase activity on NC vs CC with ref. 8? I think the authors should check the helicase activity again using the conventional gel electrophoresis helicase assay used by ref.8 for direct comparison.

Other minor concerns:

1. In the crystal structure of the complex, the structure of ASCC3 contains only residues 1-186. So in Figure 1e, the red tail in the diagram should indicate this instead of covering the tail to the 207 residue. The magenta linker is not shown in the diagram of Fig.1e as well.
2. In Fig.3d & e, the ITC titration shows ASCC3 NTR ratio of 0.8 different from 1.0 for ASCC31-197. How many replicates were the ITC titration performed? What is mean and S.D for the ratio and K_d?
3. In Fig. 3i & j, ASCC3 variants lacking 15, 41, 206 or 400 N-terminal residues was co-precipitated 73, 40, 26 or 16 %, respectively. These are not completely in agreement with their structural and mutagenesis analyses.
4. In page 10, line 205-206, it is confusing that "results show that, while dispensable for a stable interaction, the C-arm of ASCC31-207 contributes to a stable interaction with ASCC21-434,"
5. What does "residue exchanges" mean? It should be residue mutations or substitution.

Reviewer #2 (Remarks to the Author):

The manuscript by Jia, et al ("The interaction of DNA repair factors ASCC2 and ASCC3 is affected by somatic cancer mutations") presents a crystal structure and mutagenic analysis of the ASCC2-ASCC3 interaction. These proteins are part of a DNA damage repair pathway. The study provides a molecular basis for several mutations in the ASCC2-ASCC3 interface that are associated with cancer. The ASCC3 protein contains two helicase cassettes. While previous reports indicated that the N-terminal was inactive, the authors demonstrate that the N-terminal cassette does have helicase activity and that activity is auto-inhibited by the NTR region, similar (though not identical) to the homologous Brr2 RNA helicase. The results are novel and represent an important contribution to the field.

The experiments are clearly presented and appear to be sound. The crystal structure is properly refined. The manuscript is well written.

Question: Can the plug-like domain (described at the end of the results section) be added back in trans- (possibly w/o ASCC2) to test the effect on ASCC3 helicase activity? This may be a relatively straightforward experiment (unless the plug domain is insoluble when expressed by itself) and would more directly implicate the plug domain in autoinhibition of helicase activity.

Minor comment:

Figure 5D. The 3'-oh and 5'-oh (overhang) labeling is easily confused with a 3'-OH and 5'-OH (hydroxyl). I suggest altering the label to avoid confusion.

Response to Reviewer Comments

Reviewer comments are repeated in bold italics, responses are in regular font, changed text passages are highlighted in yellow.

Reviewer #1

Jia et al report the interactions of DNA repair factors ASCC2 and ASCC3 through structural, ITC and mutagenesis. They determined a crystal structure of the ASCC21-434-ASCC31-207 complex was determined via SAD approach at 2.7 Å resolution to Rwork/Rfree values of 20.4/24.7 %. Their structural analysis suggested the Interfaces between ASCC3 and ASCC2 are evolutionarily conserved. In addition, they demonstrated that the N-terminal arm of ASCC3 is essential for stable binding to ASCC2 by mutagenesis and ITC. Finally, they found that some key amino acid at the interface between ASCC2 and ASCC3 are mutated in human cancers. Overall the structure is good and the interaction data are solid.

We thank the reviewer for considering our results as good and solid.

However, there are two serious concerns:

1. The authors found that ASCC2 does not influence the helicase activity of ASCC3. If so, what is the connection between the mutations at the interface and cancers?

We thank the reviewer for pointing out the necessity to better explain how somatic cancer mutations affecting the ASCC2-ASCC3 interaction may influence DNA alkylation repair and thus cancer. While based on our findings, ASCC2 does not directly influence the helicase activity of ASCC3, we suggest that reduced ASCC2-ASCC3 affinity (as a consequence of certain somatic cancer mutations) may have a similar effect as ASCC1 knockdown, which has been shown to reduce ASCC2-ASCC3 co-localization at nuclear foci, leading to increased alkylation sensitivity.

In the first version of the manuscript, we had included a section entitled “Weakened ASCC2-ASCC3 interaction as a molecular disease principle” in the Discussion. We have now amended this section to clarify the point raised by the reviewer (lines 388-395):

Specifically, ASCC1 appears to promote co-localization of ASCC2 and ASCC3 at nuclear foci during alkylation damage, and loss of ASCC1 leads to increased alkylation sensitivity.¹⁰ Proper co-localization of ASCC2 and ASCC3, therefore, appears to be essential for efficient DNA alkylation damage repair. Similar to ASCC1 knockdown, reduced ASCC2-ASCC3 affinity, as elicited by several somatic cancer mutations, may influence ASCC2-ASCC3 co-localization at nuclear foci, leading to increased DNA alkylation damage. This model does not invoke a direct influence of ASCC2 on ASCC3 helicase activity, consistent with our findings.

2. The authors found the NC of ASCC3 has the 3'-5' helicase activity in contrast with the published results that the CC of ASCC3 has the same activity. Could ASCC2 influences the CC activity instead? Can the authors explain why they got conflicting helicase activity on NC vs CC with ref. 8? I think the authors should check the helicase activity again using the conventional gel electrophoresis helicase assay used by ref.8 for direct comparison.

As suggested, to further test our findings we now also conducted gel-based unwinding assays using a subset of our ASCC3 constructs. Fully consistent with our stopped-flow/fluorescence-based unwinding data, the gel-based assays show that

- ASCC3^{HR} is an active helicase.
- ASCC3^{NC} is an active helicase but exhibits reduced helicase activity compared to ASCC3^{HR}.
- An ASCC3^{HR} variant bearing a D611A residue substitution affecting motif II of the NC exhibits strongly reduced helicase activity.
- In contrast, the helicase activity of an ASCC3^{HR} variant bearing a D1453A residue substitution affecting motif II of the CC is not reduced compared to WT ASCC3^{HR}.

We briefly mention these additional results in the revised manuscript (lines 274-276 and lines 290-291):

Surprisingly, however, in our hands, ASCC3^{NC} also exhibited helicase activity, albeit reduced compared to ASCC3^{HR} (Fig. 5f). These findings were confirmed using gel-based unwinding assays (Supplementary Fig. S2).

Again, gel-based unwinding assays using D611A and D1453A variants of ASCC3^{HR} were consistent with these results (Supplementary Fig. S2).

We have included a description of the assay in the Methods section (lines 536-555):

Gel-based helicase assays

The same 3'-overhang DNA as for stopped-flow/fluorescence-based unwinding assays, but lacking fluorophore and quencher moieties, was used for gel-based unwinding assays (for sequences see Supplementary Table S2). The long strand was 5'-[³²P] labeled by incubating 5 μ l of 20 μ M oligonucleotide, 20 μ l of γ [³²P]-ATP (Hartmann Analytic), 3 μ l 10x reaction buffer (New England Biolabs) and 2 μ l T4 polynucleotide kinase (New England Biolabs) for 1 h at 37 $^{\circ}$ C. The labeled oligonucleotide was further passed through a Microspin G25 column (Sigma), PCI-extracted and annealed to the short, unlabeled complementary oligonucleotide in a 1:1 molar ratio in annealing buffer (10 mM TRIS-HCl, pH 8.5). Unwinding reactions were carried out at 30 $^{\circ}$ C. 150 nM purified recombinant ASCC3 variants were mixed with 12 nM DNA substrate in 40 mM TRIS-HCl, pH 7.5, 0.5 mM MgCl₂, 187 mM NaCl, 3.75 % glycerol, 0.75 mM DTT. After a 10 min incubation at 30 $^{\circ}$ C, reactions were started by addition of 5 mM ATP/MgCl₂. Aliquots were withdrawn at selected time points and reactions were quenched with 50 mM TRIS-HCl, pH 7.5, 50 mM EDTA, 0.5 % [w/v] SDS). For 0 min time points, all reactants except ATP/MgCl₂ were added. Samples were separated on 12 % non-denaturing polyacrylamide gels, DNA bands were visualized using a phosphorimager and quantified with ImageQuant software (Cytiva). The fraction of unwound DNA in each sample was calculated as $I^{ss} / (I^{ss} + I^{ds})$, in which I^{ss} is the intensity of the single-stranded DNA band and I^{ds} is the intensity of the double-stranded DNA band. The fraction unwound at 0 min was subtracted from the fraction unwound for each sample.

We also provide an additional supplementary figure (new Extended Data Fig. 2), showing the results:

Supplementary Fig. S2: Gel-based unwinding assay.

a, Gels showing time courses of unwinding of a 3'-overhang DNA substrate (long strand radiolabeled; same sequence as used in stopped-flow/fluorescence-based unwinding assays; Supplementary Table S2) by the ASCC3 variants indicated on the right. Bands representing double-stranded (ds) and single-stranded (ss) DNA are labeled on the left. The DNA input contained a mixture of dsDNA and ssDNA. Boiled, DNA sample heated to 95 °C for 5 minutes before gel analysis. **b**, Quantification of the data shown in (a). Data represent means \pm SD of three independent experiments.

We cannot explain why we obtained conflicting results for ASCC3^{NC} compared to ref 8. As we did not manage to produce isolated ASCC3^{CC} in soluble form, we more carefully phrased our conclusions regarding helicase activity of ASCC3^{NC} and ASCC3^{CC} in the revised manuscript (lines 295-299):

As the ASCC3^{NC} helicase activity was reduced compared to that of ASCC3^{HR} (Fig. 5f; Supplementary Fig. S2) it remains to be seen whether the C-terminal cassette is inactive but stimulates helicase activity of ASCC3^{NC}, as has been observed for Brr2³², or whether ASCC3^{CC} is an active helicase as well that contributes to the overall activity of ASCC3^{HR}.

Other minor concerns:

1. In the crystal structure of the complex, the structure of ASCC3 contains only residues 1-186. So in Figure 1e, the red tail in the diagram should indicate this instead of covering the tail to the 207 residue. The magenta linker is not shown in the diagram of Fig.1e as well.

We thank the reviewer for pointing this out and have adjusted Fig. 1e accordingly. Here is the revised Fig. 1e:

We explain in the updated legend to Fig. 1e:

Violet and purple bars in the ASCC2¹⁻⁴³⁴ scheme represent helices h3a/b and h6a/b. The C-terminal 21 residues of ASCC3¹⁻²⁰⁷ (gray line) are not defined in the electron density.

2. In Fig.3d & e, the ITC titration shows ASCC3 NTR ratio of 0.8 different from 1.0 for ASCC31-197. How many replicates were the ITC titration performed? What is mean and SD for the ratio and K_d?

We thank the reviewer for pointing this out and we have reassessed our ITC analysis strategy accordingly. Some deviations from the true stoichiometry are not unusual in ITC experiments. Although we took great care to prepare all proteins used to the same level of purity and to properly assess the protein concentrations, slight errors in the estimated protein concentrations could account for the observed differences. In addition, we cannot fully exclude that a small fraction of molecules in a particular preparation are non-interacting, e.g. due to some misfolding. For example, as can be seen from Fig. 1a, ASCC3^{NTR} tends to slightly degrade during cell opening and purification, presumably due to a high degree of intrinsic disorder in some regions. We did our best to limit such degradation by working swiftly and with the samples always at 4 °C, but some amount of degraded protein could not be avoided. Thus, we believe that the most likely explanation for the deviations of the estimated stoichiometries from the expected 1:1 ratio for some ITC runs is an overestimation of the concentration of interaction-active ASCC3 variants. Therefore, we have re-analyzed our data, fixing the stoichiometries at a value of 1 (which we know from our crystal structure analysis) and allowing the estimated concentrations of ASCC3 variants or ASCC3 peptides (the least well defined parameters in these experiments) to vary. The results are very similar to the outcomes of the previous analyses. We adjusted the description in the Methods section accordingly (lines 494-496):

For data analysis, stoichiometries were fixed to 1, as known from our crystal structure analysis, and the concentrations of ASCC3 variants or peptides were allowed to vary.

We now repeated all ITC runs with similar outcomes and added K_d ± SD to the revised Figs. 3 and 4.

3. In Fig. 3i & j, ASCC3 variants lacking 15, 41, 206 or 400 N-terminal residues was co-precipitated 73, 40, 26 or 16 %, respectively. These are not completely in agreement with their structural and mutagenesis analyses.

We thank the reviewer for this comment and apologize for not having discussed the results in sufficient detail. The pulldown assay does not resolve, in which exact complex ASCC2 is pulled down *via* ASCC3 variants. Additional proteins, such as other ASCC subunits or AlkBH3, are most likely pulled down concomitantly. Thus, while our structural and mutagenesis data suggest that truncation of the N-terminal 206 or 400 residues of ASCC3 would abrogate direct binary ASCC2-ASCC3 interactions, such N-terminal ASCC3 truncations still pulled down reduced amounts of ASCC2 most likely because an indirect interaction is still maintained *via* other proteins. We provided a brief explanation in the revised manuscript (lines 206-210):

While based on our structural and mutagenesis data truncation of the N-terminal 206 or 400 residues of ASCC3 should abrogate direct ASCC2-ASCC3 interactions, ASCC3^{207-end} and ASCC3^{401-end} most likely still pulled down reduced amounts of ASCC2 (Fig. 3i,j), because other ASCC subunits also mediate indirect ASCC2-ASCC3 interactions *in vivo*.

4. In page 10, line 205-206, it is confusing that "results show that, while dispensable for a stable interaction, the C-arm of ASCC31-207 contributes to a stable interaction with ASCC21-434,"

We apologize for the unclear description. What we wanted to say is that while the C-arm contributes to the overall affinity, it is not essential for a stable interaction between the proteins in SEC. Such a situation is not uncommon in protein-protein interactions that are maintained *via* several, independent contact sites – even in the absence of one interacting region, the remaining contact sites can maintain a stable complex in SEC. We adjusted the description in the revised manuscript (lines 193-195):

The C-arm of ASCC3¹⁻²⁰⁷ contributes to the interaction with ASCC2¹⁻⁴³⁴ but it is not essential for the proteins to maintain a stable complex in SEC, consistent with a reduced but still high level of evolutionary conservation of the C-arm.

5. What does "residue exchanges" mean? It should be residue mutations or substitution.

Referring to proteins bearing "residue exchanges", we wanted to describe protein variants, which due to mutations in cancer cells or due to our genetic manipulations contained amino acid residues at specific positions different from the WT versions of the proteins. We would like to avoid the expression "mutation" for describing changes on the protein level, as in our view "mutation" refers to alterations on the DNA level. We thus replaced "residue exchange(s)" with "residue substitution(s)" throughout the revised manuscript, as suggested.

Reviewer #2

The manuscript by Jia, et al ("The interaction of DNA repair factors ASCC2 and ASCC3 is affected by somatic cancer mutations") presents a crystal structure and mutagenic analysis of the ASCC2-ASCC3 interaction. These proteins are part of a DNA damage repair pathway. The study provides a molecular basis for several mutations in the ASCC2-ASCC3 interface that are associated with cancer. The ASCC3 protein contains two helicase cassettes. While previous reports indicated that the N-terminal was inactive, the authors demonstrate that the N-terminal cassette does have helicase activity and that activity is auto-inhibited by the NTR region, similar (though not

identical) to the homologous Brr2 RNA helicase. The results are novel and represent an important contribution to the field. The experiments are clearly presented and appear to be sound. The crystal structure is properly refined. The manuscript is well written.

We thank the reviewer for the very positive overall evaluation of our work and for considering our results important.

Question:

Can the plug-like domain (described at the end of the results section) be added back in trans- (possibly w/o ASCC2) to test the effect on ASCC3 helicase activity? This may be a relatively straightforward experiment (unless the plug domain is insoluble when expressed by itself) and would more directly implicate the plug domain in autoinhibition of helicase activity.

We thank the reviewer for the suggestion and performed the corresponding experiment. We compared helicase activities of ASCC3^{HR} alone or in the presence of ASCC3^{NTR} (residues 1-400) or ASCC3⁴²⁻¹⁶¹, which both contain the plug-like domain (residues 59-142). We did not observe any significant impact on the helicase activity of ASCC3^{HR} by adding the NTR or the shorter plug domain-containing fragment. Thus, a covalent connection of the NTR to the helicase region is required to elicit an effect. We have included these additional results in the revised manuscript (lines 318-321):

When added in *trans*, ASCC3^{NTR} or the plug domain-containing fragment, ASCC3⁴²⁻¹⁶¹, did not alter ASCC3^{HR} helicase activity (Fig. 5h), indicating that the NTR has to be covalently connected to the helicase region to elicit the auto-inhibitory effect.

In the Discussion (line 366-367):

Still, our data suggest that a covalent connection of the plug-like domain to the helicase region is required for auto-inhibition.

The additional results are now shown in new Fig. 5h. As the three curves are largely overlapping, we also show them side-by-side as insets in the Figure:

h, Stopped-flow/fluorescence-based assays monitoring unwinding of a 3'-overhang DNA by ASCC3^{HR} alone or in the presence of ASCC3^{NTR} or ASCC3⁴²⁻¹⁶¹ using ATP. Insets, side-by-side presentation of the largely overlapping curves.

Minor comment:

Figure 5D. The 3'-oh and 5'-oh (overhang) labeling is easily confused with a 3'-OH and 5'-OH (hydroxyl). I suggest altering the label to avoid confusion.

We agree and thank the reviewer for pointing this out. We have now chosen "3'/5'-ovh" as abbreviations.

REVIEWERS' COMMENTS

Reviewer #1 (Remarks to the Author):

The revision has satisfied my previous concerns. Therefore, this manuscript is good for publication.

Reviewer #2 (Remarks to the Author):

The authors have fully addressed my previous concerns with the manuscript.